# Species Identification of Caviar Based on Multiple DNA Barcoding

**DOI:** 10.3390/molecules28135046

**Published:** 2023-06-28

**Authors:** Qingqing Hu, Yingqiu Pan, Huili Xia, Kexin Yu, Yian Yao, Feng Guan

**Affiliations:** 1College of Life Sciences, China Jiliang University, Hangzhou 310018, China; 2Taizhou Food and Drug Inspection and Research Institute, Taizhou 318000, China

**Keywords:** DNA barcoding, caviar, *COI* gene, *D-loop* gene, species identification, authenticity identification

## Abstract

This study aimed to explore the applicability of DNA barcoding for assessing the authenticity of caviar on the Chinese market. A set of universal *COI* primers and two sets of designed primers based on *COI* and *D-loop* genes were used to identify maternal species of samples from 21 batches of caviar. The results showed that the PCR products from three sets of primers had more than 98% similarity to the sequences in database. The *COI* gene could not distinguish sturgeons with closed genetic relationships, while *D-loop* gene could effectively improve the accuracy of DNA barcoding and was more suitable to the identification of interspecific sturgeon than the *COI* gene. The neighbor-joining dendrogram further confirmed the applicability and accuracy of *COI* and *D-loop* genes in identifying maternal relatives of caviar (*Acipenser baerii*/*Acipenser gueldenstaedtii*/*Acipenser schrenckii*/*Huso dauricus*/*Huso huso)*. Despite the limitations of mitochondrial DNA in identifying hybrid sturgeon species, the presence of counterfeit caviar of non-sturgeon ingredients could be excluded. All the caviar samples were identified successfully as sturgeon species, but the mislabeling rate of species was 33.4%, indicating that there were illegal phenomena such as disorderly labeling, mislabeling, and adulteration on the market.

## 1. Introduction

Caviar is recognized as the top delicacy in the world and is commonly known as “black gold” in China; it is made from the fish roe of female sturgeon through a series of complex processing procedures [1]. Generally, caviar contains various essential amino acids, fatty acids, vitamins, minerals, and other nutrients, which are popular with a wide range of consumers [2]. Sturgeon has the characteristics of late sexual maturity, low survival rate of larvae, and strict requirements for growth environment, resulting in extremely low production of fish roe. Illegal overfishing, habitat destruction, late sexual maturity, low survival rates of juvenile fish, and other factors have greatly decreased the production of wild sturgeon. Nowadays, sturgeon aquaculture has gradually replaced wild sturgeon in the world [3]. China has become the world’s leading country in sturgeon aquaculture and caviar trade, with the main sturgeon species of caviar sold on the market being *H. dauricus*♀ × *A. schrenckii*♂, *A. baerii*, *A. schrenckii*, *H. dauricus*, *H. huso,* and *A. gueldenstaedtii* [4]. The commercial value of caviar largely depends upon the sturgeon species, but also on the simplicity of farming conditions, the length of the spawning period, and the quality of caviar (The price range is about 8~180 CNY/g, with *H. huso* caviar being the most expensive) [5]. Additionally, some artificial products similar to caviar but without any sturgeon ingredients are sold on the market, including caviar substitutes, caviar imitations, and caviar simulations (The actual market price usually does not exceed 2 CNY/g). More than 38 species of non-sturgeon fish and 3 species of other animals have been used as caviar substitutes, including salmon, trout, carp, sea urchin, sea cucumber, snail, etc. [6,7,8]. Caviar imitations are made from marine fish, lobster meat mixed with seaweed, etc., as well as from plant materials (pectin, honey and buckwheat, soybean meal, kelp, etc.) with food additives such as pigment, essence, and salt, which imitate the appearance and taste of natural caviar [9,10]. The nutritional composition of natural caviar and the caviar substitutes varies significantly among different species. Domestic and international reports on the authenticity of caviar indicated that many cases of fraud, such as replacing high-value fish roe with low-value fish roe, confusing fake fish roe with real fish roe, and illegal hunting of protected sturgeon, still exist [11,12]. In order to strengthen the regulation of the market for caviar, the development of identification technology is of great significance for the economy and food safety.

Since DeSalle and Birstein et al. [13,14] first proposed that DNA molecular methods could identify sturgeon species and caviar authenticity, many relevant studies have reported that some techniques have been used to identify sturgeon purebreds, crossbreds and their caviar, such as polymerase chain reaction–restriction fragment length polymorphism (PCR-RFLP), amplified fragment length polymorphism (AFLP), microsatellite marker, and DNA barcoding [15,16,17,18,19,20]. DNA barcoding has become a widely and commonly used technology in the identification of fish species, playing an important role in the identification of maternal species in sturgeon, and various mitochondrial markers are well-established for identifying sturgeon and caviar. However, neither the commonly used *COI* gene nor the cytochrome b (*Cytb*) gene could distinguish between *A. baerii*, *A. gueldenstaedtii*, *Acipenser naccari* and *Acipenser persicus*, while the *D-loop* gene has been found useful in discriminating between closely related species [4,16,21,22,23].

In this study, commonly used DNA barcoding primers targeted to *COI* gene [24] for fish identification were selected, excluding the presence of non-sturgeon species, and two sets of primers targeted to *COI* and *D-loop* genes were designed to specifically identify sturgeon species. Three sets of primers were analyzed to further compare and select suitable primers for the identification of caviar in order to understand the authenticity and labeling compliance of caviar products produced and sold in China, and to provide some data for market regulation of caviar products.

## 2. Results

### 2.1. Preliminary Morphological Identification

The results (Table 1) of the morphological characteristics showed that the interior of the sample was sticky and that the surface layer and original structure of samples S4, S5, and S13 were damaged, causing them to be unidentifiable. Moreover, samples S10 and S21 could be tentatively identified as *H. huso* based on the larger diameter of fish roe (about 3.3 mm or more) and its iron grey or pearl grey color. In terms of taste and flavor, samples S10 and S21 had a rich buttery fragrance, which was significantly different from that of other sturgeon products. Furthermore, the fish roe of *H. dauricus* had a similar color to those of *A. schrenckii*, which was brownish yellow or brownish gray, but the fish roe of the former was large and had a unique aroma of rich cream. The morphological characteristics of representative samples of six different caviar products are shown in Figure 1.

### 2.2. DNA Concentration and Quality

Concentrations of all the DNA samples were measured at the optical density of A260/280 nm, which was between 1.71 and 2.13 (mean 1.76 ± 0.48). PCR amplifications were performed using a *18S rRNA* primer. The 140-bp length product was consistent with the expected size, indicating that the extracted DNA met the requirements of PCR amplification (Figure 2).

### 2.3. PCR Specificity

The mixture of sturgeon DNA samples served as a positive control. Ten animal and plant DNA samples (salmon, trout, herring, cod, snail, honey, pectin, soybean, kelp, and buckwheat) [9,10] were amplified to verify the specificity of primer sets II and III. The results showed that the two self-designed primer sets had no obvious nonspecific amplification in all the DNA samples except sturgeon (Figure 3).

### 2.4. Plant-Derived Ingredients Identifying

All the caviar DNA samples were amplified using the *tRNA-Leu* primers for endogenous genes of plant. The electrophoresis results showed that no positive PCR products were obtained, indicating no plant-derived ingredients in the sampled fish roe. Therefore, the possibility of plant ingredients in caviar samples was ruled out (Figure 4).

### 2.5. Sturgeon-Derived Ingredients Identifying

All DNA samples were amplified using three primer sets (I–III). The results of electrophoresis showed that primer set I could amplify and obtain a target fragment length of ~680 bp, and primer sets II and III obtained fragment lengths of ~583 and ~520 bp, respectively, which were in accordance with the expected sizes (Figure 5). All the PCR products had no nonspecific amplification. Finally, the PCR products were recovered and directly bidirectional sequenced in Tsingke Biotechnology Co., Ltd. (Hangzhou, China).

The sequencing results obtained from six parallel experiments in the 126 samples were compared and analyzed. The sampled DNA sequences from the same batch were highly consistent and could be identified as the same species. The comparison and analysis of the data using the NCBI and BOLD databases showed that the similarity of each sample with the database was ≥98%, and the matching degree was high. The primers (set I) and self-designed *COI* gene primers (set II) achieved the same results of species identification, but neither could distinguish between *A. baerii*, *A. gueldenstaedtii*, *A. naccari*, *A. persicus*, and *A. sinensis* (Samples S1, S7, S11, S14, S19), which was consistent with the results of the previous reports [20,23]. In contrast, *D-loop* gene primer (set III) completely identified the sturgeon maternal information of all caviar samples, including *A. baerii*, *A. gueldenstaedtii*, *A. schrenckii*, *H. dauricus,* and *H. huso*; with the majority of samples being identified as *H. dauricus* (52.4%) (Table 2).

According to the Codex Alimentarius Standard “Sturgeon Caviar” and the Made in Zhejiang Group Standard “Sturgeon Caviar” [25], the labeling requirements for caviar need to indicate parentage information, especially for hybrid caviar. Seven caviar samples (S1/S3/S4/S5/S8/S16/S17) were found to be counterfeit as the maternal information did not match the product label (33.4% of the total samples), indicating that cases of fraud still exist in the Chinese market.

### 2.6. Phylogenetic Tree Construction and Analysis

The phylogenetic tree of the *COI* gene (Figure 6) and the *D-loop* gene (Figure 7) were constructed, where experimental samples were represented by “S-number.” In the *COI* gene phylogenetic tree, sturgeons were generally divided into two branches: Atlantic species and Pacific species. The Atlantic species included *A. baerii, A. gueldenstaedtii*, *A. naccari*, *A. persicus*, etc., and the Pacific species included *H. dauricus*, *A. schrenckii*, *A. sinensis*, *A. transmontanu*, etc. [26]. The clustering analysis showed that the classification results of the phylogenetic tree were highly consistent with those of DNA barcoding identification, which verified the reliability of DNA barcoding for the identification of sturgeon. However, samples S7, S14, and S19 were sequenced with the primer set I and identified as *A. gueldenstaedtii*, *A. naccari*, *A. persicus,* or *A. sinensis. A. sinensis* was not located on the same branch as the other sturgeons (Figure 6). This confusion came from the erroneous deposition in Genbank of a complete mitochondrial DNA (mtDNA) genome of *A. gueldenstaedtii* under the species name *A. sinensis*. Dillman et al. [27] used DNA sequence data from four mtDNA loci (*COI*, *Cytb*, *12S,* and *16S rDNA*) and discovered that the mitogenome for *A. sinensis*, the Chinese sturgeon, may have been either incorrectly identified or the result of undocumented hybridization with *A. gueldenstaedtii*. In addition, the result of the *COI* sequence alignment of samples S1 and S11 was *A. baerii* or *A. gueldenstaedtii*, while the species was located on the branch of *A. baerii*. Obviously, it was not reliable to distinguish related species based on the *COI* gene. However, these samples were accurately identified after amplification with primer III, and the identification results were consistent with the branches of the phylogenetic tree, ensuring the reliability of the identification results (Figure 7). Comparison of the final results of the molecular identification with those of the morphological identification revealed that the morphological species identification of S3 and S18 was manually misclassified in Table 1. Obviously, the traditional morphological identification methods are subjective and need to use DNA barcoding to ensure the accuracy of identification results. In this study, it was sensitive enough to distinguish the sturgeon-related species based on the *D-loop* gene as the target, and the *COI* gene could be used as an auxiliary target to improve the reliability and accuracy of sturgeon species identification. Further, the phylogenetic tree validated and supported the taxonomic view. It was verified that *A. gueldenstaedtii*, *A. baerii, A. naccari*, and *A. persicus* were closely-related species, and they were located on the same branch in dendrograms [4,18]. *Huso* was of non-monophyletic origin, including *H. huso* and *H. dauricus*, which were not on the same branch in the phylogenetic tree [28,29].

## 3. Discussion

Traditional morphological identification methods are not applicable to caviar that has lost its morphological characteristics, and the possibility that a product labeled as purebred is a hybrid species cannot be ruled out. So, it is necessary to use molecular identification technology to identify the caviar’s source.

DNA barcoding has been widely used for adulteration, false labeling, and raw material traceability of fish products. Mitochondrial DNA, as the main DNA barcoding for animal classification, has the advantages of more copies than others compared with nuclear genes [30,31]. This study used plant specific DNA barcodes (*tRNA-Leu*) [32] and fish universal primers (*COI*) [24] for identifying 21 batches of caviar, which ruled out the possibility of counterfeit caviar with non-sturgeon and plant-derived ingredients at the first step. Although *COI*-based DNA barcoding were suitable for detecting a wide range of animal species, several studies have shown limitations in the identification of sturgeon relatives [20,23] and we further confirmed this view. However, we successfully identified the maternal source of all caviar using primer set III based on the *D-loop* gene. Recently, the commercial frauds of caviar-substitutes have become more diverse than in the past. In this study, a high rate of mislabeling (33.4%) has been detected in 21 batches of caviar samples, only in the identification of the maternal source. Most of the mislabeled products were identified as *H. dauricus*, which does not exclude the possibility that a hybrid sturgeon caviar with *H. dauricus* as the maternity could have replaced the purebred caviar. Xia et al. [4] identified 40 kinds of caviar on the Chinese market and found a mislabeling rate of 42.5%, with the maternal species being *H. dauricus* as the majority of samples. It was suggested that there were frequent cases of counterfeit caviar on the Chinese market. There were also serious fraudulent phenomena in other countries (such as southeastern Europe), with fraudulent products featuring mislabeling, counterfeiting, and illegal fishing reaching 62.9% (17/27) [12]. There are many reasons for these behaviors. Firstly, financial gain is the greatest motivation for fraud. The market tends to substitute low-value caviar for high-value caviar. Secondly, sturgeon species, which are easily farmed and produced in large quantities, tend to be the main substitutes for fraudulent practices. According to the analysis of the current situation of sturgeon aquaculture, hybrid sturgeons (*H. dauricus*♀ × *A. schrenckii*♂ and *A. baerii* × *A. schlenckii*) have higher production performance, such as faster growth, disease resistance, and higher-quality, compared to their parents [17]. The two hybrid sturgeons mentioned above are likely to be common substitutes for other species of high-value caviar on the market, particularly *H. dauricus*♀ × *A. schrenckii*♂, and have become the main species of caviar products [33].

China attaches great importance to the market regulation and food safety of caviar. It is necessary for regulatory authorities to develop standardized molecular technology to identify caviar. In the past decade, researchers have used multiple microsatellites and SNP-sites for the identification of hybrid sturgeon and have combined them with DNA barcoding to achieve identification of parental information for the hybrid sturgeon [20,34]. In this study, the mitochondrial DNA was first selected for preliminary identification of caviar and, despite only being able to identify the maternal source, it was effectively applied to exclude the presence of heterologous substitutions and to assist the other methods to identify hybrids. These results provided important technical support for identifying caviar and increasing the database of DNA barcoding.

## 4. Materials and Methods

### 4.1. Sample Collection

A total of 21 batches of caviar products were collected by way of purchase and gift. Most of the caviar sold on the market was individually packed in iron boxes (10 g per box). All the product labels indicated the origin, raw materials, sturgeon species, and other information (Table 3). At the same time, ten species of animal and plant materials that could be used to make artificial caviar were collected as controls, including salmon, trout, herring, cod, snail, honey, pectin, soybean, kelp, and buckwheat. Except for the starch stored at room temperature, all the others were stored at −20 °C.

### 4.2. Morphological Identification

Three fish roe were randomly but discontinuously selected from each package of caviar and placed in a clean and hygienic environment with sufficient light. Their morphological characteristics were analyzed by the naked eye and a microscope, combined with the 16 sensory attributes reported by Baker et al. [35], such as color, pigment distribution, fish roe size, food flavor, and so on. The fish roe of *H. huso* and *H. dauricus* are slightly larger (fish roe diameter about 3.2 mm or more) compared with that of others; fish roe of sturgeon is elastic, has a specific flavor, has a bitter or other peculiar smell, and has no earthy taste [36]. A preliminary determination of the authenticity of caviar was established based on information from the morphological characteristics of caviar.

### 4.3. DNA Extraction

A total of 6 fish roe were randomly sampled from each independent package to rule out the possibility that the raw materials of caviar came from different sturgeon species or other fish. Sampled fish roe were washed with phosphate-buffered saline (PBS) to remove the excess preservatives two to three times repeatedly before DNA extraction. Each fish roe was thoroughly crushed to a powder and placed in a 1.5-mL centrifuge tube. DNA was extracted from the single fish roe using the Animal Tissue DNA Kit (Simgen Biotechnologies, Hangzhou, China) according to the manufacturer’s protocols.

The concentration and purity of extracted DNA were determined using an ultraviolet-visible spectrophotometer NanoDrop 2000 (Thermo Fisher Scientific, Shanghai, China), and its quality was further verified using *18S rRNA* gene primers [37]. DNA samples were stored at 4 °C for next use.

### 4.4. Primers and PCR Amplifications

The *COI* universal primers were selected and used for species identification of fish roe [24]. Mitochondrial sequences of five sturgeon species (NC012576.1, NC017603.1, NC021757.1, NC023837.1, and NC005252.1) were downloaded from the National Center for Biotechnology Information (NCBI) database. The sequences of *COI* and *D-loop* genes were aligned and analyzed by MegAlign and the CLC Genomics Workbench 8.5 for regions that were conserved and had significant differences in the precinct sequences. Sequence alignment is shown in Appendix A. The two sets of primers (sets II and III) were designed using Primer 5.0 software and tested for specificity and universality using BLAST (https://blast.ncbi.nlm.nih.gov/Blast.cgi, accessed on 20 June 2021). The transfer RNA–leucine (*tRNA-Leu*) [32] primers were selected as the control for plant ingredient identification which could sensitively detect plant-derived ingredients in processed foods and exclude adulterated caviar with plant ingredients. All primers were synthesized by Tsingke Biotechnology Co., Ltd. (Hangzhou, China) and diluted to 10 μM for use. The primers’ information is shown in Table 4.

The 25 µL PCR reaction mixtures contained 1.0 µL of each primer, 2 µL (10×) reaction buffer (MgCl_2_ free) (TaKaRa Bio, Beijing, China), 2 µL (2.5 mM) of MgCl_2_ (25 mM, TaKaRa Bio), 0.4 µL (2 U) of *Taq*-Purple DNA polymerase (5 U/µL, TaKaRa Bio), 1.2 µL (2.5 mM) of mixed dNTPs (10 mM, Simgen), and 4 µL of DNA template. Amplifications were performed under the following PCR conditions: 95 °C denaturation for 5 min, followed by 30–35 cycles; 94 °C denaturation for 35 s, corresponding annealing temperature (Table 4) for 40 s, and extension at 72 °C for 1 min; and a final extension at 72 °C for 10 min. All PCR products were inspected on 1.5% agarose gel, and GelRed was used as a stained dye.

### 4.5. Specificity Test for Designed Primers

Ten species of animal and plant DNA samples were amplified, and double-distilled water was used as the negative control to test the specificity of primer sets Ⅱ and Ⅲ. The DNA samples included salmon, trout, herring, cod, snail, honey, pectin, soybean, kelp, and buckwheat, and three parallel tests were carried out. PCR products were visualized as described above.

### 4.6. Sequencing and Analysis

PCR products were purified using an Ultra-thin DNA purification kit (Simgen Biotechnologies, Hangzhou, China) according to manufacturer’s instructions and confirmed by Sanger sequencing on the ABI 3500 sequencer using a DNA Cycle Sequencing Kit (Tsingke Biotechnology, Hangzhou, China). The obtained sequences were manually proofread using DNAStar.Lasergene.v7.1. Then, the sequences were compared and analyzed using the NCBI [38] (https://www.ncbi.nlm.nih.gov/, accessed on 20 May 2023) database and BOLD Systems [39] (http://www.boldsystems.org/, accessed on 20 May 2023) to determine species based on similarity, coverage, and overall scores. When the similarity was ≥98%, the species was determined. The sequences obtained from *COI* and *D-loop* genes were compared with 12 species of 2 genera in *Acipenseriformes* (10 species of Acipenser and 2 species of Huso) downloaded from the GenBank database to further verify the accuracy of caviar species identification using DNA barcoding, and the neighbor-joining (NJ) dendrograms were constructed based on *K-2-P* model using MEGA 5.0 [40]; the results were used to further verify the monogenic and cluster relationships of the species.

## 5. Conclusions

Based on the DNA barcoding, we designed and optimized the primers of *COI* and *D-loop* genes, effectively identifying the common maternal source of caviar on the Chinese market. Despite excluding fraudulent counterfeiting of natural caviar with bionic caviar or caviar substitutes in these samples, there is still substitution of low-value caviar for high-value caviar, causing serious fraud to consumers. Developing methods for identifying authenticity and strengthening the market regulatory systems are important measures to safeguard the legitimate trade of caviar products, protect the rights and interests of distributors and consumers, and have important social and economic significance.

## Figures and Tables

**Figure 1 molecules-28-05046-f001:**
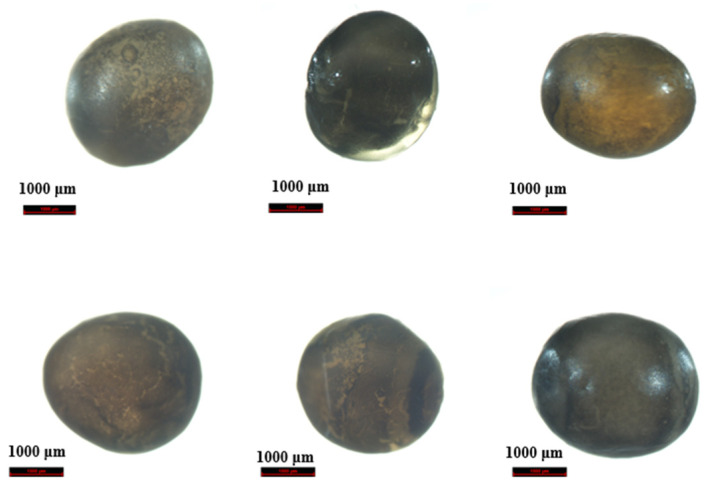
Morphological photos of some caviar products. The scale is 1000 μm; From left to right and from top to bottom the caviar is *A. gueldenstaedtiii*, *A. baerii*, *A. schrenckii*, *H. dauricus*, *A. dauricus* × *A. schrenckii*, and *H. huso*.

**Figure 2 molecules-28-05046-f002:**
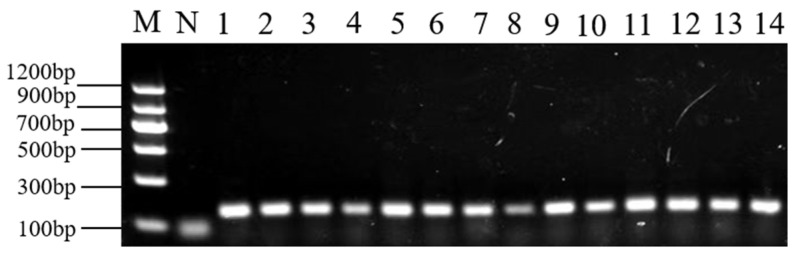
*18S rRNA* primer’s PCR results. Lane N, Negative control; Lanes 1–14, part of DNA samples.

**Figure 3 molecules-28-05046-f003:**
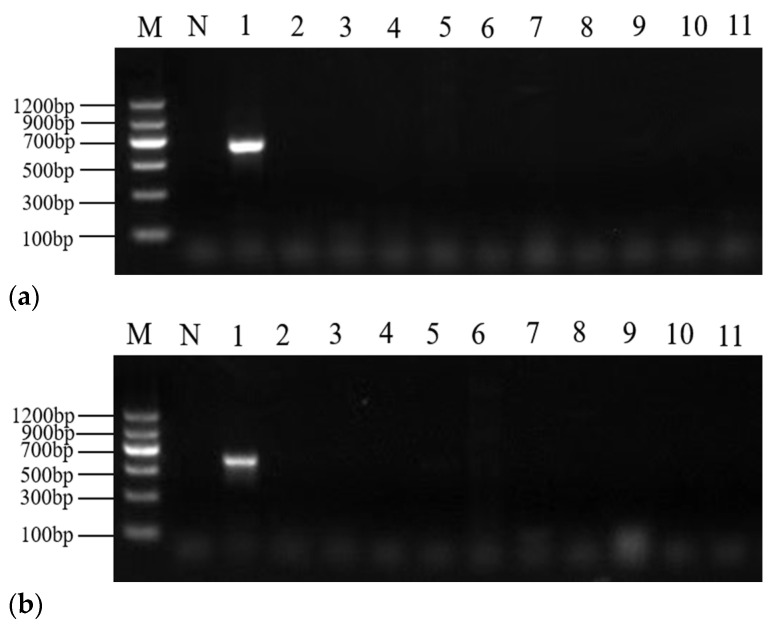
Results of specificity test for *COI* (**a**) and *D-loop* (**b**) gene designed primer set II and set III. Lane M, DNA marker; N, negative control; Lane 1, positive control; Lane 2, salmon; Lane 3, trout; Lane 4, herring; Lane 5, cod; Lane 6, snail; Lane 7, honey; Lane 8, pectin; Lane 9, soybean; Lane 10, kelp; Lane 11, buckwheat.

**Figure 4 molecules-28-05046-f004:**
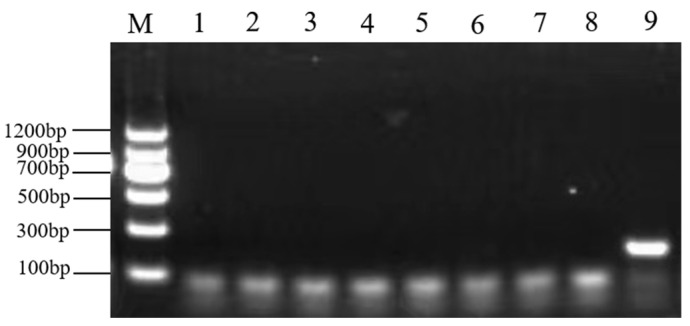
Plant component detection in caviar samples. Lane M, DNA marker; Lanes 1–8, part of caviar samples; Lane 9, positive control.

**Figure 5 molecules-28-05046-f005:**
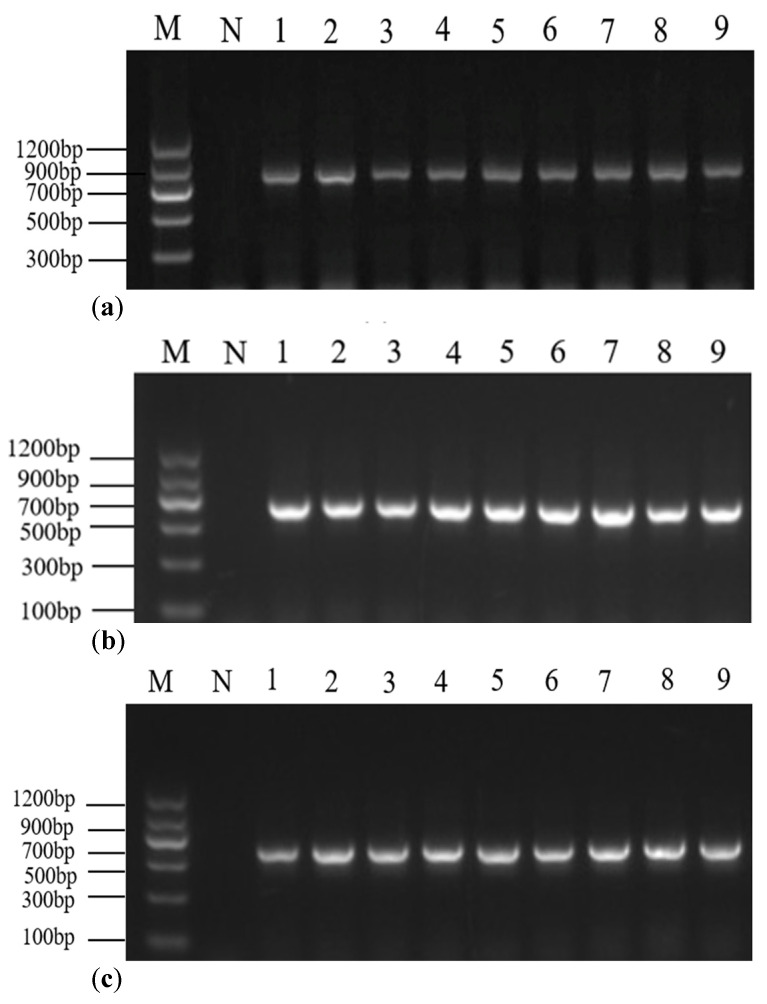
Electrophoresis of primer set I (**a**), set II (**b**), and set III (**c**) for caviar samples. Lane M, DNA marker; lane N, negative control; lanes 1 and 6, Highbury Caviar; lane 2, *H. dauricus* Caviar; lanes 3 and 5, *A. baerii* Caviar; lanes 4 and 7, *A. gueldenstaedtii* Caviar; lane 8, *H. huso* Caviar; lane 9, *A. schrenckii* Caviar.

**Figure 6 molecules-28-05046-f006:**
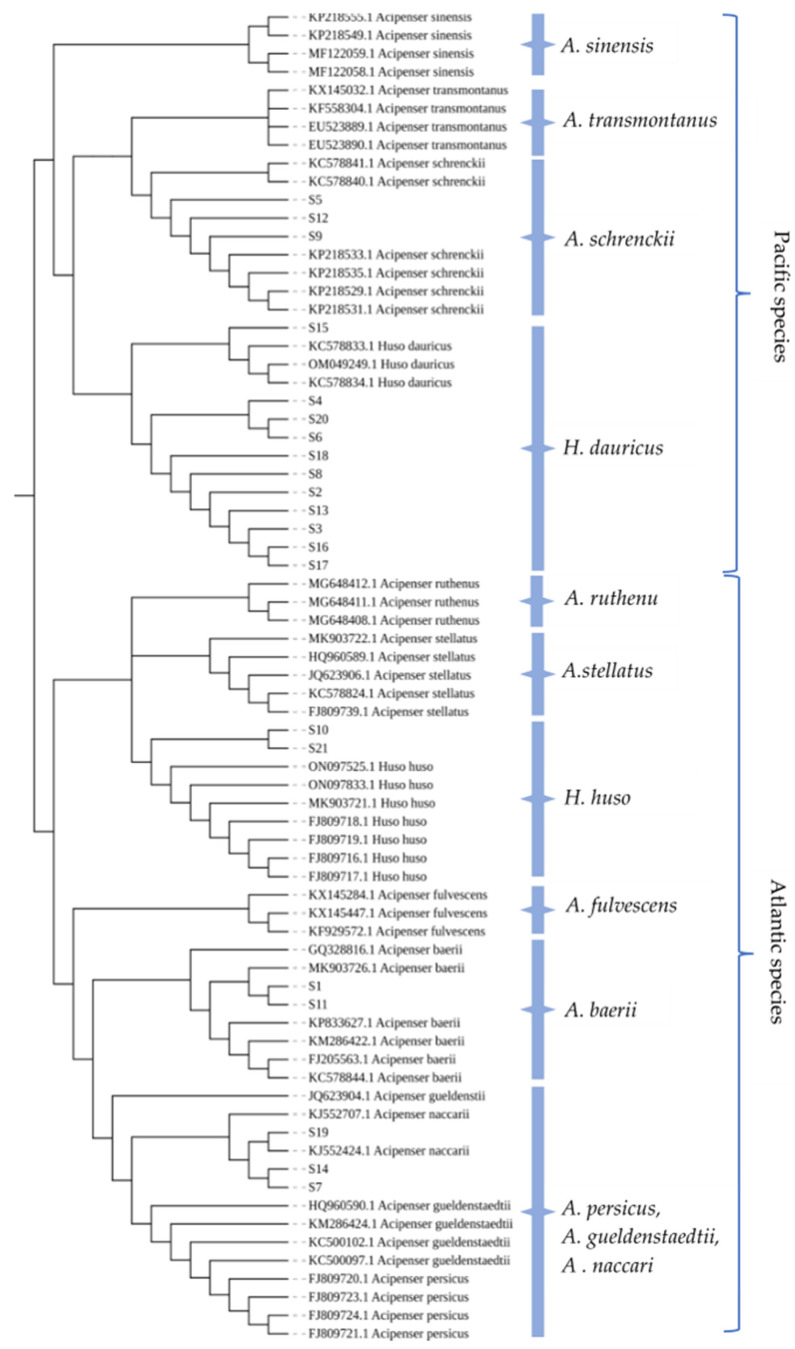
NJ phylogenetic tree using *COI* genes for some *Acipenseridae* fish and caviar samples.

**Figure 7 molecules-28-05046-f007:**
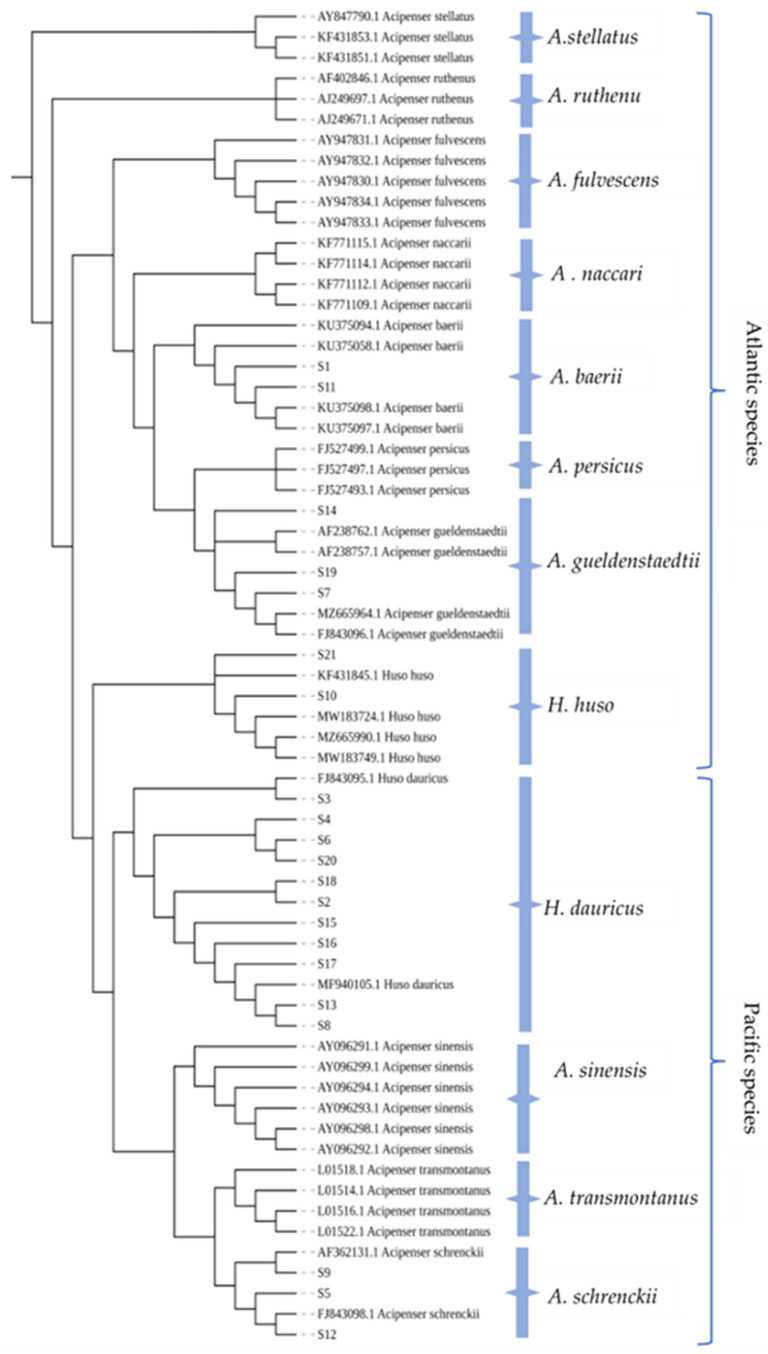
NJ phylogenetic tree using *D-loop* genes for some *Acipenseridae* fish and caviar samples.

**Table 1 molecules-28-05046-t001:** Morphological and molecular identification of caviar.

No.	Labeled Species	Morphological Identification	Molecular Identification (Maternal)
S1	*H. dauricus* × *A. schrenckii*	*A. baerii*	*A. baerii*
S2	*H. dauricus*	*H. dauricus*	*H. dauricus*
S3	*A. baerii*	*A. gueldenstaedtii*	*H. dauricus*
S4	*A. gueldenstaedtii*	/	*H. dauricus*
S5	*A. baerii*	/	*A. schrenckii*
S6	*H. dauricus* × *A. schrenckii*	*H. dauricus* × *A. schrenckii*	*H. dauricus*
S7	*A. gueldenstaedtii*	*A. gueldenstaedtii*	*A. gueldenstaedtii*
S8	*H. huso*	*H. dauricus*	*H. dauricus*
S9	*A. schrenckii*	*A. schrenckii*	*A. schrenckii*
S10	*H. huso*	*H. huso*	*H. huso*
S11	*A. baerii*	*A. baerii*	*A. baerii*
S12	*A. schrenckii*	*A. schrenckii*	*A. schrenckii*
S13	*H. dauricus* × *A. schrenckii*	/	*H. dauricus*
S14	*A. gueldenstaedtii*	*A. gueldenstaedtii*	*A. gueldenstaedtii*
S15	*H. dauricus*	*H. dauricus*	*H. dauricus*
S16	*A. baerii*	*H. dauricus × A. schrenckii*	*H. dauricus*
S17	*A. schrenckii*	*H. dauricus*	*H. dauricus*
S18	*H. dauricus* × *A. schrenckii*	*A. schrenckii*	*H. dauricus*
S19	*A. gueldenstaedtii*	*A. gueldenstaedtii*	*A. gueldenstaedtii*
S20	*H. dauricus*	*H. dauricus*	*H. dauricus*
S21	*H. huso*	*H. huso*	*H. huso*

‘/’ represents not determined.

**Table 2 molecules-28-05046-t002:** Molecular identification of maternal information of caviar.

No.	Trade Name	NCBI Database			BLOD Database	
		*D-loop* Gene (Primer Set III)	*COI* Gene (Primer Set I)	*COI* Gene (Primer Set II)	*COI* Gene (Primer Set I)	*COI* Gene (Primer Set II)
S1 *	Highbury Caviar	*A. baerii* (100%)	*A. baerii*/*A. gueldenstaedtii* (100%)	*A. baerii*/*A. gueldenstaedtii* (99.83%)	*A. baerii* (100%)	*A. baerii* (99.79%)
S2	*H. dauricus* Caviar	*H. dauricus* (99.81%)	*H. dauricus* (99.84%)	*H. dauricus* (99.58%)	*H. dauricus* (100%)	*H. dauricus* (99.79%)
S3 *	*A. baerii* Caviar	*H. dauricus* (99.81%)	*H. dauricus* (99.84%)	*H. dauricus* (99.79%)	*H. dauricus* (100%)	*H. dauricus* (99.79%)
S4 *	*A. gueldenstaedtii* Caviar	*H. dauricus* (99.81%)	*H. dauricus* (99.69%)	*H. dauricus* (100%)	*H. dauricus* (99.85%)	*H. dauricus* (100%)
S5 *	*A. baerii* Caviar	*A. schrenckii* (99.80%)	*A. schrenckii* (100%)	*A. schrenckii* (99.79%)	*A. schrenckii* (100%)	*A. schrenckii* (99.81%)
S6	Highbury Caviar	*H. dauricus* (99.81%)	*H. dauricus* (100%)	*H. dauricus* (100%)	*H. dauricus* (99.85%)	*H. dauricus* (100%)
S7	*A. gueldenstaedtii* Caviar	*A. gueldenstaedtii* (100%)	*A. gueldenstaedtii*/*A. naccari/A. persicus/A. sinensis* (100%)	*A. gueldenstaedtii*/*A. persicus*/*A. naccari* (99.83%) *A. sinensis* (99.83%)	*A. gueldenstaedtii*/*A. naccari*/*A. persicus/A. sinensis* (99.79%)	*A. gueldenstaedtii/A. naccari*(100%)
S8 *	*H. huso* Caviar	*H. dauricus* (100%)	*H. dauricus* (99.84%)	*H. dauricus* (99.79%)	*H. dauricus* (100%)	*H. dauricus* (99.79%)
S9	*A. schrenckii* Caviar	*A. schrenckii* (99.81%)	*A. schrenckii* (99.86%)	*A. schrenckii* (99.58%)	*A. schrenckii* (99.86%)	*A. schrenckii* (99.63%)
S10	*H. huso* Caviar	*H. huso* (99.78%)	*H. huso* (99.69%)	*H. huso* (99.65%)	*H. huso* (100%)	*H. huso* (99.65%)
S11	*A. baerii* Caviar	*A. baerii* (99.78%)	*A. baerii*/*A. gueldenstaedtii* (100%)	*A. baerii*/*A. gueldenstaedtii* (99.82%)	*A. baerii* (100%)	*A. baerii* (99.79%)
S12	8-year Caviar	*A. schrenckii* (99.43%)	*A. schrenckii* (99.79%)	*A. schrenckii* (99.79%)	*A. schrenckii* (99.79%)	*A. schrenckii* (100%)
S13	9-year Caviar	*H. dauricus* (99.81%)	*H. dauricus* (98%)	*H. dauricus* (98%)	*H. dauricus* (100%)	*H. dauricus* (99.79%)
S14	10-year Caviar	*A. gueldenstaedtii* (100%)	*A. gueldenstaedtii*/*A. naccari*/*A. persicus/A. sinensis* (99.79%)	*A. gueldenstaedtii*/*A. naccari*/*A. persicus*/*A. sinensis* (99.83%)	*A. naccari* (99.79%)	*A. gueldenstaedtii* (99.79%)
S15	15-year Caviar	*H. dauricus* (99.62%)	*H. dauricus* (99.79%)	*H. dauricus* (99.79%)	*H. dauricus* (100%)	*H. dauricus* (99.79%)
S16 *	*A. baerii* Caviar	*H. dauricus* (99.81%)	*H. dauricus* (99.79%)	*H. dauricus* (99.79%)	*H. dauricus* (100%)	*H. dauricus* (99.79%)
S17 *	*A. schrenckii* Caviar	*H. dauricus* (99.81%)	*H. dauricus* (100%)	*H. dauricus* (100%)	*H. dauricus* (100%)	*H. dauricus* (100%)
S18	Highbury Caviar	*H. dauricus* (99.81%)	*H. dauricus* (100%)	*H. dauricus* (100%)	*H. dauricus* (100%)	*H. dauricus* (100%)
S19	*A. gueldenstaedtii* Caviar	*A. gueldenstaedtii* (100%)	*A. gueldenstaedtii*/*A. naccari*/*A. persicus*/*A. sinensis* (100%)	*A. gueldenstaedtii*/*A. naccari*/*A. persicus*/*A. sinensis* (100%)	*A. naccari* (100%)	*A. gueldenstaedtii*/*A. naccari* (100%)
S20	*H. dauricus* Caviar	*H. dauricus* (99.81%)	*H. dauricus* (100%)	*H. dauricus* (99.79%)	*H. dauricus* (100%)	*H. dauricus* (99.79%)
S21	*H. huso* Caviar	*H. huso* (99.81%)	*H. huso* (99.69%)	*H. huso* (99.65%)	*H. huso* (100%)	*H. huso* (99.65%)

“*”: mislabeled samples; The gene sequence obtained from sequencing in this study has been uploaded to NCBI: D-loop (GenBank ID: OR095798–OR095818), COI (primer set I) (GenBank ID: OR101709–OR101729) and COI (primer set II) (GenBank ID: OR101824–OR101844).

**Table 3 molecules-28-05046-t003:** Information on Caviar.

No.	Caviar	Labeled Species
S1	Highbury	*H. dauricus* ♀ × *A. schrenckii*♂
S2	H dauricus Caviar	*H. dauricus*
S3	*A. baerii* Caviar	*A. baerii*
S4	*A. gueldenstaedtii* Caviar	*A. gueldenstaedtii*
S5	*A. baerii* Caviar	*A. baerii*
S6	Highbury Caviar	*H. dauricus* ♀ × *A. schrenckii*♂
S7	*A. gueldenstaedtii* Caviar	*A. gueldenstaedtii*
S8	*H. huso* Caviar	*H. huso*
S9	*A. schrenckii* Caviar	*A. schrenckii*
S10	*H. huso* Caviar	*H. huso*
S11	*A. baerii* Caviar	*A. baerii*
S12	8-year Caviar	*A. schrenckii*
S13	9-year Caviar	*H. dauricus* ♀ × *A. schrenckii*♂
S14	10-year Caviar	*A. gueldenstaedtii*
S15	15-year Caviar	*H. dauricus*
S16	*A. baerii* Caviar	*A. baerii*
S17	*A. schrenckii* Caviar	*A. schrenckii*
S18	Highbury Caviar	*H. dauricus* ♀ × *A. schrenckii*♂
S19	*A. gueldenstaedtii* Caviar	*A. gueldenstaedtii*
S20	*H. dauricus* Caviar	*H. dauricus*
S21	*H. huso* Caviar	*H. huso*

**Table 4 molecules-28-05046-t004:** Information on primers.

Primer Category	Primer Name	Primer Sequence (from 5′ to 3′)	Product Size (bp)	Annealing Temperature (°C)	Primer References
*18S rRNA*	18S140F	TCTGCCCTATCAACTTTCGATGG	140	56	[37]
	18S140R	TAATTTGCGCGCCTGCTG			
Set I (*COI*)	Fish F2-t1	**TGTAAAACGACGGCCAGT** CGACTAATCATAAAGATATCGGCAC	680	54	[24]
	Fish R2-t1	**CAGGAAACAGCTATGAC** ACTTCAGGGTGACCGAAGAATCAGAA			
	F2-t1	**TGTAAAACGACGGCCAGT** CAACCAACCACAAAGACATTGGCAC			
	FR1d-t1	**CAGGAAACAGCTATGAC** ACCTCAGGGTGTCC			
Set II (*COI*)	Stur-*COI*F	**TGTAAAACGACGGCCAGT** ACTGACTRGTSCCCCTAAT	583	56	Design of this study
	Stur-*COI*R	**CAGGAAACAGCTATGAC** CTATGTARCCAAAAGGTTC			
Set III (*D-loop*)	SturDF	**TGTAAAACGACGGCCAGT** ATGTARTAAGAGCCGAACA	520	54	Design of this study
	SturDR	**CAGGAAACAGCTATGAC** AGTCAGTCCTGCTTTTGG			
*tRNA-Leu*	tRNAleu-F	CGAAATCGGTAGACGCTACG	180	60	[32]
	tRNAleu-R	TTCCATTGAGTCTCTGCACCT			

Bold letters in the primer sequences represent the part of the tailed primers, that is, the sequencing primers of M13.

## Data Availability

The data presented in this study are available from the corresponding author upon reasonable request.

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
