# Peer review of "Species Identification of Caviar Based on Multiple DNA Barcoding"

_molecules, 2023, doi:10.3390/molecules28135046_

Round 1

Reviewer 1 Report

This study aims to test the DNA barcoding approach to evaluate fraud in caviar Chinese market.  The study is well designed but there is no particular novelty besides some sets of primer design. 

Line 99: Why did you choose that species? Please add citations

Line 174: add citation

Line 261: How did you design the couple of primers? Please explain and add to the text. Also would be useful if you also add a figure with the alignment (maybe in supplementary materials)

Line 262: Why did you choose tRNA-Leu for plant detection and not the common markers for plant such as rbcl, matk, ITS etc?

In table 2 add the reference for primers and indicate the primer designed in this study

Add a section in material and methods where do you explain how to construct phylogenetic analysis (remove this info from results section)

Compare your rate of fraud with the one in literature

Author Response

Dear reviewer,

We are very grateful to your comments for the manuscript. According with your advice, we tried our best to amend the relevant part and made some changes in the manuscript. These changes will not influence the content and framework of the paper. All of your questions were answered below.

Yours Sincerely.

Qingqing Hu

China Jiliang University

Reviewer 2 Report

Comments on “Species identification of caviar based on multiple DNA barcoding”

Sturgeon caviar is an expensive and luxury food product, and, because it is difficult or even impossible to attribute to the species by taste and appearance, it is often is a subject to mislabeling and fraud. Correct species designation on every tin with caviar is required both by CITES convention and by national low in many countries, including China. Authors obtained 63 bathes of caviar throughout market in China and found that 21 of them (33%) are mislabeled.

The methods authors use (mtDNA sequencing) is grossly outdated because 1. Half of the marketed caviar produced from hybrids, which proper identification can not be done by mtDNA only and require nuclear loci to be included in analysis, and 2. group of closely related species (gueldenstaedtii, naccarii, persicus) possess haplotypes, which is usually not possible to distinguish by BLAST, but need to determine particular haplotype. Also, there is confusing similarity between “baerii-like” haplotype of A. gueldenstaedtii and typical A. baerii.

D-loop is known as the most polymorphic region in sturgeon mtDNA, and it was shown 15 years ago that only this region can distinguish “baerii-like” from A baerii, while COI and many other regions are identical for Russian and Siberian sturgeon. Primers (AHR3 and LporF) and snip position to distinguish these two species, as well as a simple PCR-based test system is provided in Mugue et al., 2008 (Polymorphism of the mitochondrial DNA control region in eight sturgeon species and development of a system for DNA-based species identification)

There are plenty of recent papers providing utility of various nuclear loci to determine paternal origin of hybrids, including STR loci (Barmintseva et al., 2013) and SNPs ( Ogden et al., 2013, Boscari et al., 2014, Havelka et al., 2019 and few other), and test for hybrid origin of caviar is routine now worldwide.

Another flaw of the manuscript is controversy about A. sinensis. By phylogenetic tree provided by authors, A.sinensis is on Pacific branch of the tree, while baerii, gueldenstaedtii, naccarii are the part of atlantic clade of sturgeon phylogeny. However, authors state

“but neither could distinguish between A. baerii, A. gueldenstaedti, A. naccari,  A. persicus, and A. sinensis (Samples S1, S7, S11, S14, S19), which was consistent with the 136 results of the previous reports “ (line 135-136)

This confusion came from the erroneous deposition in Genbank of complete mtDNA genome of A. gueldenstaedtii under the species name A. sinensis. This was found a while ago and this case is described in “Forensic investigations into a GenBank anomaly: endangered taxa and the importance of voucher specimens in molecular studies, Dillman et al.,2014” Soon after the correct mitogenome of A. sinensis was obtained and deposited in Genbank, but the mislabelled mitogenome often appears in the top results when anyone blasts gueldenstaedtii mtDNA.

Also I would recommend not to use part of D-loop with 82bp repeats because common heteroplasmy in this region (Ludwig et al., 2000 (Heteroplasmy in the mtDNA control region of sturgeon (Acipenser, Huso and Scaphirhynchus), Wang et al., 2009 (Heteroplasmy in mtDNA control region and phylogenetics of five sturgeons). Primers developed in the manuscript target this region makes sequencing very unreliable.  

In summary – I would not present results of caviar identification without proper analysis of nuclear loci, or, at least, I would recommend to highlight throughout title, abstract and results section that current report cover only maternal origin of the caviar tested.

Author Response

(The authors gave the same response as above.)

Reviewer 3 Report

The general idea is not new but still interesting. I have some concerns regarging the specifity of primers, I mentioned that in my comments in pdf file. Were the samples amplified in replications? Did you take enough safety prcedures to separate the positive and negative controls? There is no information if the obtained sequences from own study were deposited in any public database like GenBank? It is common practise nowdays. 

The manuscript is quite well written but needs linguistic revision. Some sentences are not clear or too long and too complicated. It should be corrected by native speaker.

Author Response

(The authors gave the same response as above.)

Round 2

Reviewer 2 Report

A had two major issues regarding the original version of the MS - limitation of mtDNA markers to detect caviar of hybrid origin and case of irroneous A. gueldenstaedtii mitogenome labelled as a A. sinensis by Genbank record. I am satisfied with changes authors made in the revised version. The main result of the presented study - the caviar on the chinease market can be successfully tested and iilgal labelling can be detected.